# The Geomagnetic Field (GMF) Modulates Nutrient Status and Lipid Metabolism during *Arabidopsis thaliana* Plant Development

**DOI:** 10.3390/plants9121729

**Published:** 2020-12-08

**Authors:** Monirul Islam, Gianpiero Vigani, Massimo E. Maffei

**Affiliations:** Plant Physiology Unit, Department of Life Sciences and Systems Biology, University of Turin, 10135 Turin, Italy; monirul.islam@unito.it (M.I.); gianpiero.vigani@unito.it (G.V.)

**Keywords:** *Arabidopsis thaliana*, Helmholtz coils, gene expression, fatty acid content, surface alkane content, micronutrients, macronutrients, near null magnetic field

## Abstract

The Geomagnetic field (GMF) is a typical component of our planet. Plant perception of the GMF implies that any magnetic field (MF) variation would induce possible metabolic changes. In this work was we assessed the role of the GMF on *Arabidopsis thaliana* Col0 mineral nutrition and lipid metabolism during plant development. We reduced the local GMF (about 40 μT) to Near Null Magnetic Field (NNMF, about 30 nT) to evaluate the effects of GMF on Arabidopsis in a time-course (from rosette to seed-set) experiment by studying the lipid content (fatty acids, FA; and surface alkanes, SA) and mineral nutrients. The expression of selected genes involved in lipid metabolism was assessed by Real-Time PCR (qPCR). A progressive increase of SA with carbon numbers between 21 and 28 was found in plants exposed to NNMF from bolting to flowering developmental stages, whereas the content of some FA significantly (*p* < 0.05) increased in rosette, bolting and seed-set developmental stages. Variations in SA composition were correlated to the differential expression of several Arabidopsis 3-ketoacyl-CoAsynthase (*KCS*) genes, including *KCS1*, *KCS5*, *KCS6*, *KCS8,* and *KCS12*, a lipid transfer protein (*LTPG1*) and a lipase (*LIP1*). Ionomic analysis showed a significant variation in some micronutrients (Fe, Co, Mn and Ni) and macronutrients (Mg, K and Ca) during plant development of plants exposed to NNMF. The results of this work show that *A. thaliana* responds to variations of the GMF which are perceived as is typical of abiotic stress responses.

## 1. Introduction

As sessile organisms, plants have evolved both constitutive and inducible responses to the changing environment. Several environmental factors affect plant growth and developments. Among them, the Earth magnetic field or geomagnetic field (GMF, about 40 μT) is an environmental component of our planet and its reduction to Near Null Magnetic Field (NNMF, about 30 nT) has been shown to influence many plant biological processes [1,2]. One of the major effects of GMF reduction is the delay in flowering time and the alteration of clock gene amplitude clock [3,4,5]. Growing plants in NNMF condition also alters the plant mineral nutrition, through modulating channels, transporters and genes involved in mineral absorption and assimilation [6,7]. Three different mechanisms of magnetoperception have been described: (1) a mechanism involving radical pairs (i.e., magnetically sensitive chemical intermediates that are formed by photoexcitation of cryptochrome [8]), which has been demonstrated in plants [9]; (2) the presence of magnetic field (MF) sensory receptors present in cells containing ferromagnetic particles, as has been shown in magnetotactic bacteria [10]; and (3) the detection of minute electric fields by electroreceptors in the ampullae of Lorenzini in elasmobranch animals [11]. Plants show both light-dependent [9,12,13] and light-independent [3,14] magnetoreception, which may reflect a differential ability of plant organs to interact with light and the GMF. Cryptochromes which are blue-light photoreceptors are involve in magnetoreception and the effect of magnetic fields was reported in cryptochrome phosphorylation [9,15,16,17]. Phytochromes, the red-light photoreceptors, are known to interact with cryptochrome at many levels and enhances and maintains plant downstream responses to blue light [18]. Phytochrome *phyAphyB* deficient mutants showed a visibly enhanced response to an applied MF [9]. Therefore, plants can use different cues to perceive variations in the MF and triggers signal transduction events eventually leading to biochemical and developmental changes.

Lipids play crucial roles in plant development being both basic components of cellular membranes and constituents of suberin and cutin waxes that provide structural barriers to the environment [19,20]. They contribute to inducible stress resistance through the remodeling of membrane fluidity [21]. Other than their structural and functional features, lipids play also important roles as signaling and regulatory molecules in cell biology. In plants, several physiological processes linked to plant growth and plant responses to environmental stimuli are under control of a complex network of lipid-mediated signaling [22]. Variations of MF have been found to alter the composition and content of radish and onion polar and neutral lipids and the composition of their fatty acids [23,24,25] affecting lipid synthesis in chloroplast, mitochondrial, and other cell membranes [26,27]. The epicuticular layers, which are mainly composed by linear and branched alkanes derived from FA elongation and decarbonilation, are involved in the external cell barrier against the entry of pathogens and the limitation of transpiration [28].

In order to investigate the role of the GMF on both lipid metabolism and growth of Arabidopsis plants during development, we exposed plants to NNMF conditions. Here we show that the reduction of the GMF to NNMF affects plant nutrition by altering the content of both micro and micronutrients and the lipid content and composition of Arabidopsis by a differential modulation of several 3-ketoacyl-CoAsynthase (*KCS*) genes, a lipid transfer protein (*LTPG1*), a lipase (*LIP1*), and a fatty acid desaturase (*ADS4*).

## 2. Results and Discussion

### 2.1. The GMF Modulates Wax Alkane Production during Plant Development

In plant exposed to NNMF we found a progressive increase of cuticular wax alkanes from the early stages of plant development (rosette) to the late phase of plant growth before senescence (seed-set), when compared to GMF (Figure 1). In general, we identified several cuticular wax alkanes with a chain length ranging from 21 to 31 carbon atoms (see material and methods for alkane nomenclature). Our results are in line with the general composition of Arabidopsis wax alkanes, with molecules ranging from 21 to 29 carbon atoms [20,29]. In particular, in the rosette stage of development (See Appendix A), exposure of plants to NNMF induced a significant (*p* < 0.05) reduction in the content of almost all identified alkanes (with the sole exception of C31) (Figure 1A). The major alkane at this stage of development was C25 in GMF exposed plants, whereas in NNMF exposed plants C26, C27 and C31 were more abundant (Figure 1A). In the bolting stage of development (Appendix A), the major alkanes were C29 and C31 in both GMF and NNMF exposed plants, which showed no quantitative significant (*p* > 0.05) differences in the content of these alkanes (Figure 1B). However, significant (*p* < 0.05) increases of C22, C24, C25, C26, C27, C28, and C30 were observed in plants exposed to NNMF with respect to GMF control plants (Figure 1B). During the flowering stage of development (Appendix A), a strong and significant increase for most of the alkanes was found in plant exposed to NNMF with respect to GMF control plants, with the sole exception for C30, which was more abundant in GMF plants (Figure 1C). Finally, in the seed-set stage of development (at 35 DAS, Appendix A), both NNMF and GMF exposed plants showed a drastic reduction of alkanes from C21 to C30 (Figure 1D). The only remaining alkanes were C29 and C31, which were significantly (*p* < 0.05) higher in NNMF plant with respect to GMF control plants (Figure 1D).

### 2.2. The GMF Modulates Fatty Acid Production during Plant Development

Several fatty acids (FAs) were affected by the reduction of the GMF, including palmitic (C16:0), palmitoleic (C16:1), stearic (C18:0), oleic (C18:1), linoleic (C18:2) and linolenic (C18:3) acids (Figure 2). In the rosette stage of development, NNMF plants showed a significantly (*p* < 0.05) higher content of C18:2 with respect to GMF plants, whereas no significant difference was found for the other FAs (Figure 2A). Almost the same results were found in the bolting stage of development, but in this case NNMF plants showed also a significant (*p* < 0.05) increase of C18:1 and a decrease of C18:0, with respect to GMF plants (Figure 2B). In the flowering stage, NNMF plants showed a significant (*p* < 0.05) reduction of C18:2 with respect to GMF plants (Figure 2C), whereas in the seed-set stage of development the C16:1 and C18:1 of NNMF exposed plants showed a significant (*p* < 0.05) increase with respect to GMF plants (Figure 2D). The increase of 18:1 and 18:2 and the reduction of 18:0 observed in our study are in line with the effects observed in other plants exposed to reduced MF [24,27].

### 2.3. The GMF Modulates the Expression of Lipid-Realted Genes during Plant Development

Among the several genes that are involved in lipid metabolism we selected some genes involved in fatty acids and alkane metabolism. Figure 3 shows the fold-change gene expression of Arabidopsis plants exposed to NNMF with respect to control plants exposed to GMF during plant development. In the rosette stage of development, upregulation of *LTPG1, KCS1,* and *KCS12* was correlated to the NNMF-dependent increased content of C18:2 (see Figure 2A), whereas the downregulation of *KCS5* and *KCS6* (Figure 3) was coherent with the reduction of alkane synthesis in NNMF exposed plants (see Figure 1A). The expression of *LTPG1* has been shown to alter the ultrastructure of the cuticle layer of the stem epidermis and to cause the reduction of C29 alkane, which is a major component of cuticular waxes in the stems and siliques [30]. While *KCS12* is known to affect wax biosynthesis [31], *KCS1* has been shown to be involved in wax biosynthesis, but with limited effect on C29 alkanes [32]. In the bolting stage, the increased content of C18:1 and C18:2 in NNMF plants (see Figure 2B) was correlated to the upregulation of *LTPG1* (Figure 3), whereas the increase of alkanes from C21 to C28 of NNMF plants (Figure 1B) was associated to the upregulation of *KCS6*, *KCS8*, *KCS12,* and *LIP1* (Figure 3). In this stage of plant development, the reduction of C29 and C31 in NNMF plants was correlated to the downregulation of *KCS5* (Figure 3). In the flowering stage, the consistent increase of alkanes from C21 to C28 in NNMF plants (see Figure 1C) was associated to the upregulation of *ADS4, KCS5*, *KCS6*, *KCS8*, *KCS12,* and *LIP1* (Figure 3). Finally, in the seed-set stage, the almost undetectable alkane from C1 to C28 observed in both GMF and NNMF (Figure 1D) might depend on to the low expression and downregulation (in NNMF condition) of *KCS1*, *KCS6*, *KCS8*, *KCS12*, and *LIP1* (Figure 3), whereas the increased content of C29 and C31 in NNMF plants (Figure 1D) correlated to the upregulation of *KCS5* (Figure 3). However, we cannot exclude that owing to the are relatively stable nature of alkanes, these changes could be due to degradation by other enzymes. *KCS6* has been correlated to changes in alkane composition during development [29] whereas the regulation of *KCS5* (also known as *CER60*) has been shown to be involved in wax biosynthesis of long-chain alkanes [33]. Our results indicate a close association between the regulation of *KCS5* and the production of C29 and C31 alkanes. *LIP1* drives the synthesis of an active triacylglycerol (TAG) lipase and is capable of hydrolyzing long chain triacylglycerol [34]. The expression of *LIP1* correlated with the synthesis of NNMF alkanes, particularly during bolting and flowering, where these wax constituents showed the highest content.

In a previous work we showed that photoreceptors, including cryptochrome, are involved in plant magnetoreception [15]. The functional modification of cryptochrome under NNMF conditions involves suppression of the phytohormone gibberellin (GA) biosynthesis [35]; and GA is involved in the modulation of fatty acid synthesis [36]. Therefore, we cannot exclude that plants may recruit signaling cascades that have evolved to respond to other stimuli and stress factors to give rise to magnetosensitive response. 

Overall, such results indicated that the GMF modulates the lipid composition and metabolism in Arabidopsis and that reduction of the GMF to NNMF impairs the plant metabolism at different developmental stages of plants. These data are in line with previous developmental studies [2,3,5]. There was not a specific trend of gene expression during plant development; for instance, expression of *KCS5* at NNMF-treated rosette and bolting stages was down-regulated, but upregulated at the flowering and seed-set stages. This might depend on the possibility that specific regulation of protein content as well as of the related enzymes activity occurs at different timing. The effect of higher induction on *KCS5* at the flowering stage might result in C29 and C31 accumulation later at the seed-set stage. Nevertheless, we observed a link between induction of gene expression and metabolites accumulation.

### 2.4. The GMF Modulates Plant Morphology

Considering that several physiological processes linked to plant growth are under control of a complex network of lipid-mediated signaling [22], we investigate the role of the GMF on the growth of Arabidopsis plants at the different developmental stages by determining morphometry and parameters of shoots under NNMF conditions (Appendix A). Under NNMF condition, the leaf area index (LAI) was significantly (*p* < 0.05) reduced in the flowering and seed set developmental stages. As expected, the shoot length was reduced in NNMF treatment from bolting to seed-set, as previously observed [5], and was accompanied by a progressive reduction of both fresh weight and dry weight (Appendix A). We also confirmed that the flowering time was delayed under NNMF conditions (see page 3 of Appendix A). The reduction of plant height under NNMF has been correlated to a delayed in flowering [2,4,5] and our results imply also an involvement of the lipid metabolism in this process.

### 2.5. The GMF Modulates Plant Homeostasis

We recently demonstrated that NNMF condition impairs nutrient homeostasis in plants [7] at the early stage of plant development. In order to evaluate the effect of the GMF during plant development, we performed an elements composition analysis of plants at the rosette, bolting, flowering and seed-set stages. 

At the rosette developmental stage, no significant variations were observed for the macronutrients, whereas iron (Fe) was the only micronutrient showing a significant (*p* < 0.05) increased when compared with GMF conditions (Table 1). This finding is in agreement with our previous observations [7] and highlights the importance of the GMF for Fe homeostasis in plants during early development.

During bolting, a significant (*p* < 0.05) increase was observed only for the micronutrient cobalt (Co). Cobalt was found to increase its availability upon treatment of barley (*Hordeum vulgare*) with magnetic nanoparticles, with a significant increase with respect to control [37]. 

While plants did not display variation in the nutritional status during the flowering stage, NNMF exposure strongly affected the content of several nutrients in leaves of plants at the seed set developmental stage. In particular, macronutrients like magnesium (Mg), potassium (K) and calcium (Ca) showed a significant (*p* < 0.05) increase in their content with respect to GMF-exposed plants. Moreover, micronutrients such as manganese (Mn), molybdenum (Mo) and nickel (Ni) showed a significantly (*p* < 0.05) higher accumulation under NNMF when compared to plants grown in GMF conditions. Interestingly, during seed set the content of Co was significantly reduced by exposure to NNMF (Table 1).

The inorganic elements content profile of leaves is considered to be a signature of the nutrient status of plants under stress conditions [38,39]. Along with morphometric data, ionomic analysis highlight that MF variations are perceived as a stress condition by plants in a developmental stage-dependent manner.

### 2.6. The GMF Modulation Highlights the Link between Lipid Metabolism and Plant Nutrition

A link between the mineral nutrient status and the lipid metabolism in higher plants has been demonstrated. Indeed, several studies reported upregulation of 3-Ketoacyl-CoA synthase genes including *KCS12*, *KC17* and *KCS21*, in *Chlamydomonas* and rice plant under N, Fe, Pi and S starvation [40,41], as well as an increase of several fatty acids such as 16:0, 16:1, 18:0, 18:1, 18:2 and 18:3 in pea leaves under Fe deficiency condition [42]. Fe excess treatment leads to alteration of lipid metabolism, with upregulation of protease inhibitor/seed storage/lipid transfer (LTP) and 3-ketoacyl-CoA synthase (KSC) proteins [43]. Therefore, under NNMF conditions, the observed increase of Fe content might be associated to the upregulation of *KCS12* and *LTPG1* found in leaves of Arabidopsis at the rosette stage. In addition, the variation of some macronutrients and micronutrients at the seed-set stage might be linked to the upregulation of *KCS5* and downregulation of *ADS4*, *LIP1* and *LTPG1* genes. In fact, a down regulation of three *KCS* genes, such as 3-ketoacyl-CoA synthase-12, -17 and -21, has been observed in Mg-deficient *Citrus sinensis* plants [44]. Furthermore, Mn-deficient Arabidopsis plants displayed upregulation of gene encoding for the fatty acid desaturase family protein ADS4 [45]. Several lipase genes were upregulated under P starvation and down regulated under K-deficient condition in rice [46] and *Citrus sinensis* leaves [44].

Additionally, Wu et. al. [47] reported that, molybdenum application can induce alteration of fatty acids of thylakoid membranes particularly increasing of C16:0, C18:2 and induce low temperature in wheat.

## 3. Materials and Methods

### 3.1. Plant Materials and Growth Conditions

*Arabidopsis thaliana* ecotype Columbia 0 (Col-0) were vernalized at 4 °C for 72 h in dark condition in order to synchronize the plants. Seeds were then sown in 8 cm diameter polyethylene pots with soil prepared with a mixture of peat and vermiculite (2 parts of peat and 1 part of vermiculite). For each developmental stage, at least 10 pots for both control and NNMF exposed plants were assayed and the experiment was repeated at least three times (see also Appendix A). Sown pots (in both GMF and NNMF exposed plants) were exposed to homogenous irradiation from a high pressure sodium lamp source (Grolux 600W, SYLVANIA, Antwerpen, Belgium) at 200 µmol m^−2^ s^−1^, 22 °C (±1.5 °C) with a photoperiod of 14 h light and 10 h darkness. Control experiments (GMF) were performed in the same lab at a distance of 6 m from the NNMF controlling system and the measured levels of power-line frequency (50 Hz), light and temperature (22 °C) of both controls and treatments were similar. Treated plants were grown inside the NNMF (see below) controlling system. All experiments were performed under normal gravity. Because NNMF delays plant flowering [5], sampling timing was set on control plants (i.e., plants exposed to GMF) and was performed at 15 days after sowing (DAS) during the early vegetative stage of development, at 21 DAS during the bolting stage of development, at 30 DAS at full bloom and after 35 DAS during the seed-set stage of plant development (see also Appendix A). For Morphological analysis, each time course pictures were collected to measure leaf area index and stem length. Leaf area index (LAI) was measured by dividing the leaf area by the pot area. Stem length was measured from the base to the tip of the flowering stem. Fresh and dry biomass were also calculated.

### 3.2. GMF Control System

The GMF B values of control plants were in line with the values of the Northern hemisphere [48]. The geographical coordinates of the laboratory were 45°0′59″ N and 7°36′58″ E and the B value measured was 40.3 µT with an inclination of +57 degrees. Near-null magnetic field (NNMF) was obtained by three couples of orthogonal Helmholtz coils connected to three direct current (DC) power supplies as previously detailed [5]. Real-time monitoring of the magnetic field inside the NNMF controlling system was achieved as previously described [5]. Polyethylene pots containing Arabidopsis seeds were exposed either to NNMF or to GMF. Double-blinded experiments were performed and a quantitative descriptions of applied fields and their spatial distributions along with measurement uncertainties were as fully described earlier [3,5].

### 3.3. Fatty Acid and Cuticular Alkane Chemical Composition

For both FA and alkane analyses and each time point, Arabidopsis plants exposed to either GMF or NNMF were harvested and immediately frozen in liquid nitrogen and the plant material was stored at −80 °C. The plant material from pooled pots was then used for extractions.

#### 3.3.1. Fatty Acid Analysis

The pooled collected leaves from each developmental stage (from 20 to 40 g, depending on the developmental stage) were extracted using cyclohexane (1:10, weight to volume ratio) and then esterified with boron tri-fluoride (10% *w*/*v* in methanol). Fifty μg heptadecanoic acid (C17:0) were added as internal standard [49]. The fatty acid methyl esters (FAME) were obtained by acid catalysis according to Christie and Han [50] and were dehydrated with anhydrous MgSO_4_. FAME identification and quantification was performed by gas chromatography-mass spectrometry (GC-MS) (5975T, Agilent Technologies, Santa Clara, CA, USA) and by gas chromatography flame ionization detector (GC-FID) (GC-2010 Plus, SHIMADZU, Kyoto, Japan), respectively, as reported earlier [49]. At least three technical replicates were run for each developmental stage of plants exposed to either GMF or NNMF.

#### 3.3.2. Cuticular Alkane Analysis

The pooled collected leaves from each developmental stage (from 20 to 40 g, depending on the developmental stage) were extracted with a mixture of 5 ml pentane:hexane (5:1) for 60 s; 30 μg of n-tritriacontane were added as internal standard. The extract was concentrated by a gentle stream of gaseous N_2_, passed through a column of anhydrous MgSO_4_ and then analysed by GC-FID and GC-MS as previously described [51]. An average of three injections was done for each sample. The following abbreviations and the diagnostic ions (*m*/*z*) of the identified alkanes were: C21 n-heneicosane (296), C22 n-docosane (310), C23 n-tricosane (324), C24 n-tetracosane (338), C25 n-pentacosane (352), C26-hexacosane (366), C27 n-heptacosane (380), C28 n-octacosane (394), C29 n-nonacosane (408), C30 n-triacontane (422), C31 n- hentriacontane (436).

### 3.4. Ionome Analysis

For ionomic analyses, leaves were collected from each time point of Arabidopsis plants exposed to either GMF or NNMF, and dried in an aerate oven at 70 °C for four days. Samples dry weight were measured and digested by using 65% HNO_3_ at 120 °C. The mineralized samples were transferred into polypropylene test tubes and diluted 1:40 in MILLI-Q water. Finally, the concentration of metal elements was measured by inductively coupled plasma–mass Spectrometry ICP-MS (BRUKER Aurora-M90 ICP-MS) as previously described [7].

### 3.5. RNA Isolation and cDNA Synthesis

For each developmental stage, 100 mg of frozen samples exposed to either GMF or NNMF were ground separately in liquid nitrogen with mortar and pestle. Total RNA was isolated using the Machery-Nagel RNA Isolation mini Kit (Machery-Nagel, Duren, Germany), and RNase-free DNAse, according to the manufacturer’s protocols. RNA quality and quantity were checked as previously described [3]. cDNA was synthesized starting from 1 µg RNA using the High Capacity cDNA Reverse Transcription kit (Applied Biosystem, Foster City, CA, USA), according to the manufacturer’s recommendations. Reaction mixtures were incubated at 25 °C for 10 min, 37 °C for 2 h, and 85 °C for 5 min.

### 3.6. Quantitative Real-Time PCR (qPCR)

All qPCR experiments were performed on a Stratagene Mx3000P Real-Time System (La Jolla, CA, USA) using SYBR green I with ROX^®^ as an internal loading standard. The reaction was performed with 25 µL of mixture consisting of 12.5 µL of 2X Maxima^TM^ SYBR Green/ROX qPCR Master Mix (Fermentas International, Burlington, ON, Canada), 0.5 µL 1:5 diluted cDNA and 100 nM primers (Integrated DNA Technologies, Coralville, IA, USA). Controls included non-RT controls (using total RNA without reverse transcription to monitor for genomic DNA contamination) and non-template controls (water template). All primers were designed using Primer 3 software [52]. Primers used for real-time PCR are reported in Appendix A. Specifically, PCR cycles were 10 min at 95 °C, 45 cycles of 15 s at 95 °C, 20 s at 57 °C, and 30 s at 72 °C and all runs were followed by a melting curve analysis with the following gradient: 1 min at 95 °C, 30 s at 55 °C, 30 s at 95 °C for *At1g06350*, *Δ9 DESATURASE 4* (*ADS4*); *At1g27950*, *GLYCOSYLPHOSPHATIDYLINOSITOL-ANCHORED LIPID PROTEIN TRANSFER 1* (*LTPG1*); *At1g01120*, *3-KETOACYL-COA SYNTHASE 1*, (*KCS1*); *At1g25450*, 3-KETOACYL-COA SYNTHASE 5 (*KCS5*); *At1g68530*, *3-KETOACYL-COA SYNTHASE 6* (*KCS6*); *At2g15090*, *3-KETOACYL-COA SYNTHASE 8* (*KCS8*); *At2g28630*, *3-KETOACYL-COA SYNTHASE 12* (*KCS12*); *At2g15230*, *LIPASE 1* (*LIP1*). Four different reference genes *At2g37620*, *ACTIN1 (ACT1)*; *At5g19510*, *ELONGATION FACTOR 1B ALPHA-SUBUNIT 2 (eEF1Balpha2)*; *At1g13440*, *CYTOPLASMIC GLYCERALDEHYDE-3-PHOSPHATE DEHYDROGENASE (GAPC2)*; and *At1g51710*, *UBIQUITIN SPECIFIC PROTEASE 6 (UBP6)*, which were used to normalize the results of the qPCR. The best performing of these four genes was selected using Normfinder software (MOMA, Aarhus, Denmark) [53]. Normfinder analysis revealed that the most stable gene was *eEF1Balpha2*.

All amplification plots were analyzed with the Mx3000P software to obtain Ct values. qPCR data are expressed as fold change of NNMF with respect to control plants growing in GMF conditions.

### 3.7. Statistical Analyses

The data obtained from lipidomics and qPCR were treated by using Systat 10 (Systat Software, San Jose, CA, USA). Mean values were calculated along with the standard deviation (SD). Paired *t* test and Bonferroni adjusted probability were used to assess the difference between treatments and the control. ANOVA was calculated in the comparative analysis of plant development data and a Tukey post hoc test was assessed.

## 4. Conclusions

Our results show that the GMF modulates *Arabidopsis thaliana* nutrition and lipid composition and this effect is related to the developmental stage of plants and that its presence is necessary for a normal development and metabolism. By reducing the GMF to NNMF we observed an increased amount of epicuticular alkanes and a modulation of FAs unsaturation. These events were associated to alteration in the nutritional status of NNMF exposed plants. Homeostasis and lipid composition are sensitive to several abiotic stresses [54,55,56] and our results suggest that the alteration of the GMF might be considered a source of abiotic stress. For instance, plants respond to varying MF by increasing reactive oxygen species (ROS) production [1,9,16,57,58], and ROS generation is a typical response to stress conditions in plants [59]. Our working hypothesis is that the GMF, just like light, gravity and touch can be perceived by plants and variations in the GMF might cause a stress condition (see also [60]). By considering that the GMF varies with latitude, we may expect variations in plants exposed to different GMF values. The evidence of differential root and shoot [57] and both light-dependent and light-independent responses to GMF variations [15,17] point to an hypothetical mechanism that might rely on different effectors. The search for intracellular effectors of GMF will surely provide more insights on plant magnetoreception.

## Figures and Tables

**Figure 1 plants-09-01729-f001:**
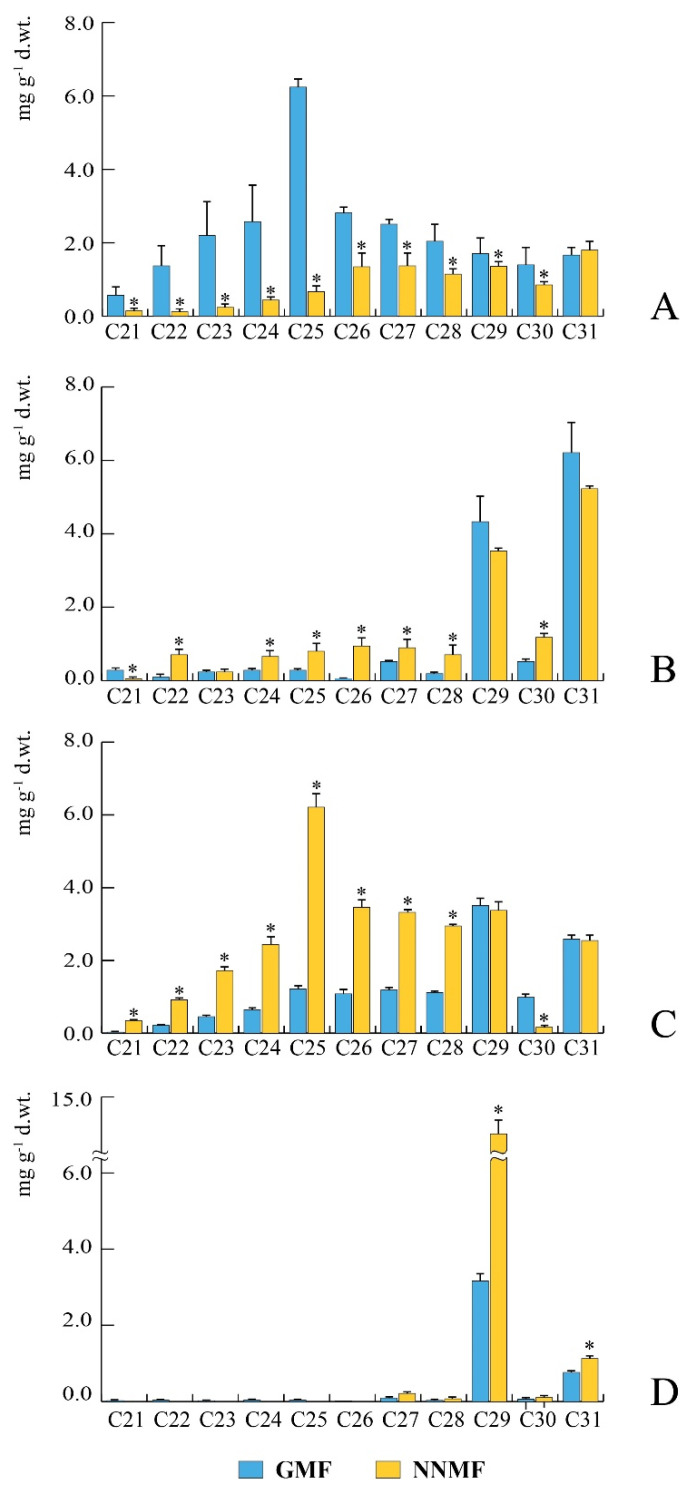
Surface alkane composition during *Arabidopsis thaliana* development upon exposure to either geomagnetic field (GMF, blue bars) or near-null magnetic field (NNMF, yellow bars). (**A**), rosette developmental stage, 15 days old plants. (**B**), bolting developmental stage, 21 days old plants. (**C**), flowering developmental stage, 30 days old plants. (**D**), seed-set developmental stage, 35 days old plants. Bars represent standard deviation. *, significant (*p* < 0.05) differences between NNMF and GMF plants.

**Figure 2 plants-09-01729-f002:**
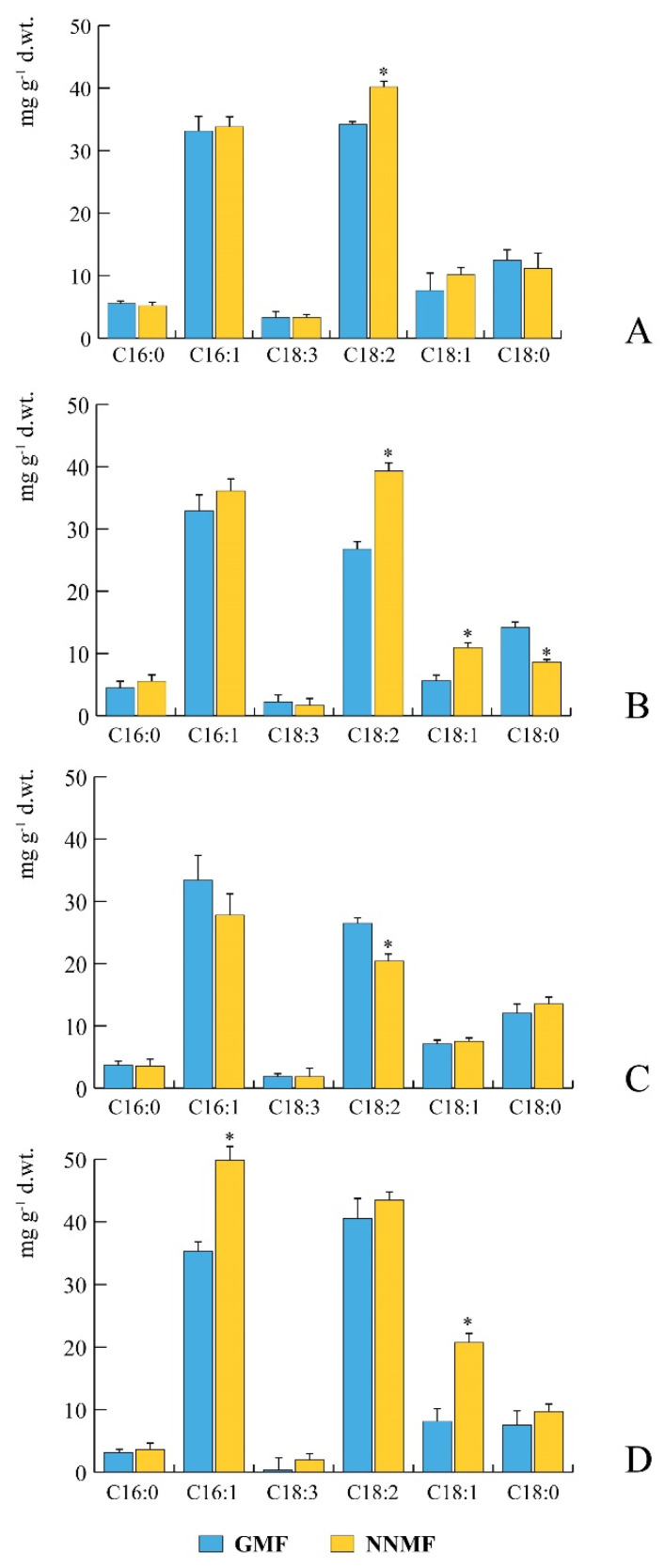
Developmental changes in fatty acid composition of *Arabidopsis thaliana* exposed to geomagnetic field (GMF, blue bars) or to near-null magnetic field (NNMF, yellow bars). (**A**), rosette developmental stage, 15 days old plants. (**B**), bolting developmental stage, 21 days old plants. (**C**), flowering developmental stage, 30 days old plants. (**D**), seed-set developmental stage, 35 days old plants. Bars represent standard deviation. *, significant (*p* < 0.05) differences between NNMF and GMF plants.

**Figure 3 plants-09-01729-f003:**
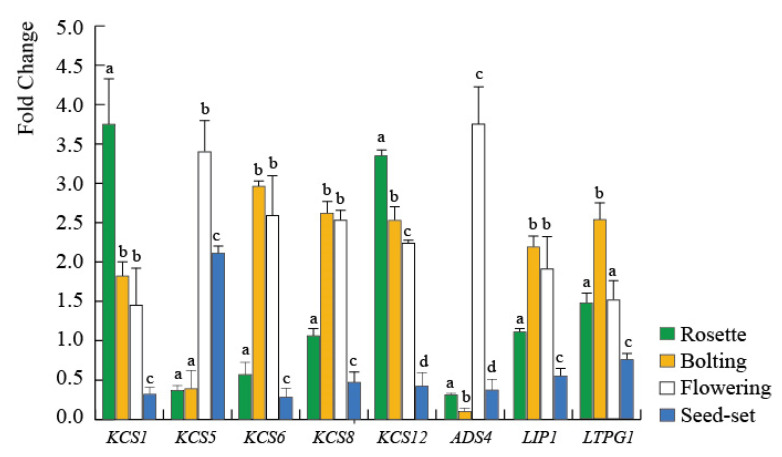
Time-course expression of genes in *Arabidopsis thaliana* exposed to NNMF conditions. Values are expressed as fold change with respect to control plants growing in GMF conditions. Bars represent standard deviation. Age of plants was the same as reported in Figure 1 and Figure 2. For the same gene, different letters indicate significant (*p* < 0.05) differences between the different plant developmental stages.

**Table 1 plants-09-01729-t001:** Macro and micronutrients contents in different developmental stages from rosette to seed-set of Arabidopsis exposed to GMF (control) and NNMF. Data are expressed as mean value (±standard deviation) from three independent biological replications. ** Boldfaced figures indicate significant (*p* < 0.05) difference between GMF and NNMF.

	Rosette		Bolting		Flowering		Seed-Set	
	GMF	NNMF	GMF	NNMF	GMF	NNMF	GMF	NNMF
Macronutrients (mg/gDW)								
Na	3.47 ± 1.67	2.11 ± 0.56	1.15 ± 0.09	1.38 ± 0.21	1.00 ± 0.87	0.64 ± 0.49	0.43 ± 0.09	1.09 ± 0.38
Mg	4.29 ± 0.34	4.04 ± 0.04	3.40 ± 0.04	3.74 ± 0.63	3.30 ± 1.43	2.36 ± 1.78	**2.71 ± 0.38**	**3.84 ± 0.48 ****
K	36.19 ± 2.46	36.76 ± 2.50	30.25 ± 3.74	31.75 ± 5.42	32.73 ± 14.79	21.93 ± 16.43	**29.61 ± 1.76**	**34.46 ± 2.33 ****
Ca	19.77 ± 1.58	20.11 ± 1.09	15.46 ± 0.99	17.28 ± 2.58	13.57 ± 6.35	9.93 ± 7.68	**8.18 ± 2.14**	**16.26 ± 3.86 ****
P	7.28 ± 0.27	7.64 ± 0.28	5.63 ± 0.44	5.62 ± 0.77	5.77 ± 2.07	4.08 ± 3.03	6.32 ± 0.43	5.23 ± 1.40
Micronutrients (µg/gDW)								
Mn	23.59 ± 1.06	27.29 ± 2.84	19.75 ± 0.70	20.14 ± 10.07	17.84 ± 4.96	10.98 ± 7.97	**18.23 ± 1.33**	**21.99 ± 1.37 ****
Fe	**334.24 ± 57.52**	**610.57 ± 153.40 ****	287.77 ± 82.51	281.50 ± 166.94	126.5 ± 69.09	112.94 ± 23.12	90.93 ± 23.33	142.47 ± 35.31
Co	0.18 ± 0.02	0.11 ± 0.09	**0.13 ± 0.04**	**0.20 ± 0.02 ****	0.07 ± 0.04	0.12 ± 0.02	**0.14 ± 0.02**	**0.07 ± 0.04 ****
Ni	3.67 ± 0.97	5.55 ± 3.53	2.31 ± 0.70	3.89 ± 2.34	2.37 ± 2.10	1.69 ± 1.15	**1.66 ± 0.54**	**3.25 ± 0.15 ****
Cu	15.02 ± 2.37	17.90 ± 5.63	14.07 ± 1.88	10.18 ± 2.29	12.70 ± 10.50	8.38 ± 6.14	9.98 ± 1.69	11.87 ± 1.12
Zn	318.15 ± 87.91	346.33 ± 267.53	228.44 ± 85.19	194.07 ± 42.44	456.97 ± 672.99	77.27 ± 62.36	76.88 ± 11.29	82.09 ± 4.83
Mo	12.63 ± 10.32	7.38 ± 2.33	3.69 ± 1.08	4.92 ± 1.84	2.93 ± 1.37	2.21 ± 1.70	**1.77 ± 0.30**	**3.31 ± 0.79 ****

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
