# Peer review of "The Geomagnetic Field (GMF) Modulates Nutrient Status and Lipid Metabolism during Arabidopsis thaliana Plant Development"

_plants, 2020, doi:10.3390/plants9121729_

Round 1
Reviewer 1 Report
The current manuscript addresses the topic of magnetic field effects on plant growth and development, which is becoming increasingly important to a wide audience due to excitement generated by the emerging field of Quantum Biology and magnetic field perception in living organisms. These authors have addressed effects at near-zero magnetic fields, which are physiologically realistic in that they occur within the range that plants may have faced in the course of their evolution. The authors document many novel changes in lipid composition, lipid content, and expression of selected genes involved in lipid metabolism as a function of magnetic field condition and plant developmental stage. The experimental setup is coherent, the controls are correctly, done, and the results provide a new insight into the complexity of magnetic field effects in plants, with potential practical applications. What is particularly important in evaluating this study, is that several past works from this lab on magnetic field effects in plants, particularly changes in gene expression related to clock, light, and hormone signaling, and cryptochrome protein phosphorylation studies, have been replicated by at least 3 independent laboratories world wide. This provides the most conclusive evidence that the protocols and methodology in place in this laboratory are valid, and addresses a concern with magnetic field studies, namely that there is sometimes controversy between results coming from different labs. In sum, given the credibility of the past work of these authors, given their careful and scientifically appropriate setup and controls , and given the wealth of new detail on effects of NNMF on significant plant physiological processes; the manuscript makes an important contribution and should be of interest to a wide audience, including those outside of the plant field.
I have the following minor comments on the discussion and interpretation of the results:
- Since the authors have looked only at long-term effects of NNMF on plant growth, they cannot pinpoint any 'initial' event that may trigger the outcome in these different pathways. However, a discussion of a possible role for cryptochrome and other photoreceptors in these processes would be appropriate, as well as of hormone signaling pathways known to be involved in control of lipid metabolism.
- It would be useful for a more general audience reading this paper if a brief synopsis of possible mechanisms of magnetic field perception could be provided (for instance magnetite-base, radical pair mechanism based), even if they cannot identify which may have been involved.
- The authors raise the intriguing point that the effect of NNMF is similar to that of abiotic stress. They should expand on this comment in the discussion (for instance explain other cases, and connection with mechanisms for magnetic field perception such as the radical pair hypothesis - see my comment no. 2 above)
- I note that there are no photographic images of plants at the different developmental stages. This is not a requirement, and not essential, if the authors have any publishable images available, it would be an added feature if the authors could provide images of each developmental plant stage they used in the experiments, comparing plants grown in presence and absence of NNMF. This makes the material used more appreciable to the general (non-plant) audience
- Finally, although the number of days (age) of plants exposed to the different (NNMF or local field) conditions may be the same, their developmental stage (eg. time to flowering) may still be somewhat affected by the magnetic field condition, for instance time to flowering is known to be altered by NNMF under certain illumination conditions. The authors should somewhere explicitly state if any plant growth stage was delayed/accelerated by NNMF.
- Finally, there are many spelling errors, a spelling check would be helpful.
Author Response
I have the following minor comments on the discussion and interpretation of the results:
- Since the authors have looked only at long-term effects of NNMF on plant growth, they cannot pinpoint any 'initial' event that may trigger the outcome in these different pathways. However, a discussion of a possible role for cryptochrome and other photoreceptors in these processes would be appropriate, as well as of hormone signaling pathways known to be involved in control of lipid metabolism.
R: We thank the reviewer for this comment. We expanded the concept of magnetoreception in both the introduction and the discussion by including the role of photoreceptors and phytohormone gibberellin in lipid metabolism.
- It would be useful for a more general audience reading this paper if a brief synopsis of possible mechanisms of magnetic field perception could be provided (for instance magnetite-base, radical pair mechanism based), even if they cannot identify which may have been involved.
R: The Introduction section has been improved by following the reviewer’s suggestion
- The authors raise the intriguing point that the effect of NNMF is similar to that of abiotic stress. They should expand on this comment in the discussion (for instance explain other cases, and connection with mechanisms for magnetic field perception such as the radical pair hypothesis - see my comment no. 2 above)
R: the concept of abiotic stress has been expanded both in the discussion and the conclusions
- I note that there are no photographic images of plants at the different developmental stages. This is not a requirement, and not essential, if the authors have any publishable images available, it would be an added feature if the authors could provide images of each developmental plant stage they used in the experiments, comparing plants grown in presence and absence of NNMF. This makes the material used more appreciable to the general (non-plant) audience
R: pictures of the different developmental stage are depicted in the supplementary material
- Finally, although the number of days (age) of plants exposed to the different (NNMF or local field) conditions may be the same, their developmental stage (eg. time to flowering) may still be somewhat affected by the magnetic field condition, for instance time to flowering is known to be altered by NNMF under certain illumination conditions. The authors should somewhere explicitly state if any plant growth stage was delayed/accelerated by NNMF.
R: we explicitly stated that the flowering time was delayed in the section related to morphological changes as suggested
- Finally, there are many spelling errors, a spelling check would be helpful.
R: the text has been completely revised
Reviewer 2 Report
Please see attached file.

Author Response
- The authors did not discuss why the effects of NNMF on lipid and nutrient contents were dependent on developmental stages.
R: we rephrased the sentence related to development as suggested
- Based on the observed effects of NNMF on lipid and nutrient contents, whether such effects of NNMF will be stable during the same developmental stage.
R: we did not perform continuous measurements but selected specific timings as reported. Therefore, we do not know whether fluctuations may be present inside the same time period. However, this is an interesting point that we surely consider for future experiments.
- The content of a certain lipid is usually determined by the balance between its biosynthesis and degradation or flux into other products. The authors only examined the expression of genes involved in biosynthesis but not other related genes involved in the metabolic pathway.
R: we agree with the reviewer. However, we mainly focused on the expression of genes involved in the lipid biosynthesis because: i) we aimed to investigate the impact of NNMF condition on the content of several lipid compounds and not just on certain lipid (which content depend on the metabolic flux balance) and ii) we considered the lipid profile during different plant growth stages which required de novo synthesis of compounds. It is known that during their growth, plants adapted to the adverse conditions through the reorganization of lipid membranes resulting from the change in the fatty-acid content and, consequently, the formation of lipids (ReszczyÅ„ska, E., Hanaka, A. Lipids Composition in Plant Membranes. Cell Biochem Biophys 78, 401–414 (2020). https://doi.org/10.1007/s12013-020-00947-w).
- In line 67, “Whereas C26 and C27 were more abundant in NNMF exposed plants (Fig. 1A)”. It seems that except for C31, all other alkanes were reduced in NNMF-treated plants.
- Lines 121-122, “Finally, in the seed-set stage, the drastic reduction of NNMF alkanes from C21 to C28 (Fig. 1D) was associated to the downregulation of KCS1, KCS6, KCS8, KCS12, and LIP1 (Fig. 3)”。Such a reduction in both NNMF and GMF should not be due to the decreased expression of these genes since alkanes are relatively stable chemicals. It could be due to that these alkanes were degraded by the related enzymes.
R: the sentence has been rephrased considering the helpful suggestions of the reviewer
- There was no detailed information about sampling. Since NNMF treatment will delay plant flowering, how the authors distinguish the effect of NNMF from that of the developmental stage on lipid components.
R: Sampling has been better described at the beginning of the materials and methods section
- The authors suggest that CS5 is highly related to the content of C29 and C31 (line 126-127). Expression of KCS5 at NNMF-treated rosette and bolting stages was down-regulated, but upregulated at the flowering and seed-set stages (Fig. 3). It was observed that C29 and C31 content was decreased and increased, respectively. However, the biggest difference in C29 content was at the seed-set stage (Fig. 1D), whereas the most difference in expression of KCS5 was at the flowering stage. How can the authors explain such a situation?
R: it is not easy to ascribe the difference in metabolite content to the different in expression of genes, since specific regulation of protein content as well as of the related enzymes activity could occur at different timing. The effect of higher induction on CS5 at flowering stage might result in C29 and C31 accumulation later at the seed-set stage. Therefore, we speculated the existence of a possible link between induction of gene expression and metabolite accumulation. Accordingly, we added a few sentences discussing this important aspect
Minors
Leaf area index (LAI) should not have a unit in supplementary Table S1. In addition, the LAI value at the seed-set stage were above 100. It seems that such a large value was impossible and also not consistent with the plants shown in Supplementary Figures.
R: we thank the reviewer for this comment and we are sorry for the mistake. We indeed corrected the LAI values as reported in table S1
Reviewer 3 Report
The authors are presenting a detailed study of the effect of near-null magnetic fields on developmental and molecular parameters of Arabidopsis thaliana. The Turin lab is in the comfortable position to possess excellent equipment, i.e. triple-coil systems, that allow the generation of precision fields even at very low magnetic-flux densities. The techniques are sound and the conclusions drawn are justified. Accept.
Minor points.
221: 3.2. GMF control system The authors indicate nowhere in this paragraph what the local magnetic-flux density and inclination angle of the “GMF” is. The geographical coordinates don’t say anything about the actual magnetic field in the lab. Buildings are shielding to some extent the geomagnetic field (about 10-12%) and affect also the inclination and polar angles. This information should be included here even though the value of 40 microT was mentioned in the Abstract. Indicating the inclination angle would be indicated, particularly, because the Turin lab showed previously that it affects development.
85: “2.2. The GMF is required for fatty acid production during development”
“Required” Would actually imply that the compounds are absent in NNMF, which is not the case. Better: affect or modulate. The manuscript is full of these somewhat misleading expressions.
300: “Our results show that the GMF is necessary for Arabidopsis thaliana nutrition and lipid composition and…” Instead of “is necessary” better “affects” or “modulates” or the like. “Necessary” would (in my understanding) imply that the response would be absent in NNMF.
Author Response
221: 3.2. GMF control system The authors indicate nowhere in this paragraph what the local magnetic-flux density and inclination angle of the “GMF” is. The geographical coordinates don’t say anything about the actual magnetic field in the lab. Buildings are shielding to some extent the geomagnetic field (about 10-12%) and affect also the inclination and polar angles. This information should be included here even though the value of 40 microT was mentioned in the Abstract. Indicating the inclination angle would be indicated, particularly, because the Turin lab showed previously that it affects development.
R: We thank very much the reviewer for noticing this. B values and the inclination have been added
85: “2.2. The GMF is required for fatty acid production during development”
“Required” Would actually imply that the compounds are absent in NNMF, which is not the case. Better: affect or modulate. The manuscript is full of these somewhat misleading expressions.
R: we made the changes accordingly and we modified the title
300: “Our results show that the GMF is necessary for Arabidopsis thaliana nutrition and lipid composition and…” Instead of “is necessary” better “affects” or “modulates” or the like. “Necessary” would (in my understanding) imply that the response would be absent in NNMF.
R: we made the changes accordingly
Round 2
Reviewer 2 Report
In the supplementary Figure 1 (Rossette stage of development), It seems that the first and third pots treated by GMF for 15 days were almost the same. Please check this problem carefully!
Author Response
We thank the reviewer for noticing this. We replaced the figures with different plants.